# Pecan Biomass and Dairy Manure Utilization: Compost Treatment and Soil In-Situ Comparisons of Selected Pecan Crop and Soil Variables

Emily F. Creegan [1],[*], Robert Flynn [2], Catherine E. Brewer [3],[*], Richard J. Heerema [1], Murali Darapuneni [1] and Ciro Velasco-Cruz [1]

1   Plant and Environmental Sciences, New Mexico State University (NMSU), Las Cruces, NM 88003, USA; rjheerem@nmsu.edu (R.J.H.); dmk07@nmsu.edu (M.D.); cvelasco@nmsu.edu (C.V.-C.)
2   Plant Extension Sciences, NMSU Agricultural Science Center, Artesia, NM 88210, USA; rflynn@nmsu.edu
3   Chemical and Materials Engineering, New Mexico State University (NMSU), Las Cruces, NM 88003, USA
*   Correspondence: ecreegan143@gmail.com (E.F.C.); cbrewer@nmsu.edu (C.E.B.)

**Abstract:** A compost program was developed on-farm, utilizing tree trimming biomass from a commercial pecan farm comprised of 14-year-old improved cultivar Western Schley pecan (*Carya illinoinensis*) tree stands. The direct soil application of shredded pecan tree biomass (P) and dairy manure (M) served as a standard on-farm practice. Three composts were produced using P and M with varying levels of other inputs and processing. The PM compost contained only P and M and its production included only weekly turning and watering. The other two composts included P, M, unfinished compost, and clay inputs, and either additional landscaping residues (A) (designated PM/A compost) or "green chop" (on-farm grown legumes, G) (designated PMG/A compost); production of PM/A and PMG/A composts included additional processing steps intended to improve compost quality per the recommendations of a compost consulting company. Soil samples were taken at three depths (0–15 cm, 15–30 cm, 30–61 cm) in November 2017 from the 1.3 ha study plot of trees. The standard practice and compost treatments were applied at approximately 18 t/ha in January 2018 and 2019 at a 15 cm depth. Soils were re-sampled at the end of the two-year study. Composts and soils were analyzed for: pH, sodium adsorption ratio (SAR), electrical conductivity, and total carbon, organic matter, magnesium, calcium, sodium, nitrate-N, total Kjeldahl nitrogen (TKN), available phosphorus, potassium, zinc, manganese, iron, and copper contents. Pecan tree leaf nutrient content, stem water potential, and leaf greenness were also measured one and two years after soil amendment application. While increases in several soil properties were observed with the treatments, only available phosphorus content was significantly different between pre and post at all depths. Electrical conductivity, TKN, Fe, Cu, SAR, and Na content showed significant differences in the upper soil layers. No differences in leaf properties were observed. This suggests that there are minimal differences in the outcomes for compost application compared to in-situ biomass application; additional compost inputs and processing did not provide additional short-term soil or plant benefits for pecan tree production. More work is needed to determine if there are long-term benefits to soil quality, plant health and performance, or carbon sequestration that impact the economic and environmental decision-making processes for composting and application of local organic wastes.

**Keywords:** *Carya illinoinensis*; organic waste; tree leaf nutrients; soil nutrients; arid regions

## 1. Introduction

Compost research often describes the effects of compost application on soil and plant properties; however, much less research has been conducted on organic waste generator-user partnerships and compost program development, particularly with agricultural waste streams [1]. There are air and water quality implications with improperly managed agricultural biomass waste streams [2]. Tree trimmings are often burned, adversely contributing

to air quality and greenhouse gas emissions. Dairy manure is often left fallow, posing deleterious effects on watersheds. This study investigated the organic waste materials from two of New Mexico's top agricultural industries—pecan and dairy [3].

The United States Natural Resources Conservation Service (NRCS) defines soil health as "the continued capacity of soil to function as a vital living ecosystem that sustains plants, animals, and humans" and cites the most critical indicator of soil health as soil organic matter (SOM) [4]. Compost is derived from organic matter and is a somewhat-mineralized and stable growing media that has been shown to improve soil physical, chemical, and biological characteristics [5,6]. "Humus" is used in soil science to describe highly degraded (physically, chemically, and biologically altered) SOM fractions that largely contribute to soil retention of moisture and nutrients [7].

Farmers are concerned with plant production, soil nutrition, and potential savings in both water and soil nutrient inputs, resource maximization, and economic gains [8]. Water conservation initiatives are typically less about addressing shortages of water and more about soil water conservation strategies to avoid losses from runoff, evaporation, and drainage [9]. Properly managed, finished compost is a safe soil amendment that builds soil structure and increases soil nutrient availability and water holding capacity [5,10]. Compared to synthetic fertilizers, compost nutrient concentrations are typically lower but are applied at much higher rates; over time, these higher-rate, lower-concentration applications can have a significant cumulative effect on soil carbon and associated soil water conservation [11,12].

Reducing the agricultural vulnerabilities associated with climate change is particularly important in productive arid regions prone to drought [13,14]. Research on perennial crops and utilization of crop residual biomass in arid regions is lacking [15]. Compost program development must show a net benefit to farm economics while meeting logistical, safety, and regulatory factors [16]. The US Composting Council asserts that correct compost processing diminishes weed seeds and reduces pathogens [17]. In contrast with composting, in-situ application of organic materials can cause nutrient immobilization and may transfer pathogens from the substrate material to the crop; this is particularly salient with ground-level crops [18].

This study was conducted to determine the employability of on-farm and locally-derived organic waste materials and compost methods and practices. As compost consulting companies continue to emerge, it is important to evaluate the production of economically-viable, quality composts. The objectives of this study were to compare the soil and plant responses of composting vs. direct soil application of pecan tree biomass and dairy manure, and simple (few material and labor inputs) vs. enhanced (more material and labor inputs) composting methods. Soil responses were compared at different depths to assess the movement and availability of salts and nutrients, which can be a concern for high initial manure applications.

## 2. Materials and Methods

### 2.1. Study Site and Standard On-Farm Practices

The experiment was conducted from May 2017 to May 2019 at a commercial pecan farm in Roswell, New Mexico (33.3943° N 104.5230° W). The average annual precipitation and temperature in Roswell are approximately 33 cm and 16 °C, respectively, and the climate is characterized as semi-arid [19]. The experimental site was comprised of approximately 1.3 ha of 14-year-old improved cultivar Western Schley pecan (*Carya illinoinensis*) trees. The site had received no prior organic material inputs and, due to the age of the trees, the trees had never been hedged. The site was flood irrigated, flowing from the west to the east. The soil on the site was identified as a Reakor loam, a fine-silty, mixed, superactive, thermic Typic Haplocalcid; the Reakor series is a deep, well-drained soil that formed in loamy alluvium [20].

As a standard practice on the farm, around 10 kg N per hectare for every 100 kg of estimated crop on the trees is typically applied in the tree crown in the form of urea (46-0-0)

and 10-30-4 liquid fertilizer. Historically, pecan trees produce 953 kg per hectare, so an average of 45.36 kg/ha·yr N is utilized. The N was distributed over four applications: before the first irrigation in March, then once each in April, May, and June. Foliar nutrient applications begin mid-April and are spaced two to three weeks apart at 1000 gallons (3.785 m$^3$) over 4 ha. Each tank contained Elite Zn 23% (3.78 L), Manniplex MN (5% Mn, 7% N, 9.5 L), Manniplex FE (5% Fe, 5% N, 9.5 L), and UAN N (32% N, 9.5 L). Trimmed tree limbs are shredded in place, and manure is added to those areas at 9 t/ha (in addition to the N fertilizer) to improve the decomposition of the high-C woody trimmings without competing with the trees for N. The shredded tree biomass and manure are rototilled or chisel ripped into the soil. If there is compost to be applied, a rate of 9 t/ha is used. As the experimental plot comprised relatively young pecan tree stands, compost was applied at a higher rate (18 t/ha).

### 2.2. Compost Production and Compost Sampling

Three composts were produced in windrows (each approximately 91 m in length and 1 m in height) using pecan tree trimming biomass from on-farm hedging (P) and local dairy cattle manure (M) with varying levels of other inputs and processing. Table 1 summarizes windrow substrates, quantities, and processing procedures. For the PM treatment, general US Composting Council guidelines were followed for compost production [17]. The PM compost contained only P and M. The PM windrow was created on 16 June 2017, turned (aerated) and watered once per week for a total of 11 h of processing time, and completed on 22 August 2017. The windrow turner (Aeromaster PT-130, Midwest Bio-Systems, Inc., Tampico, IL, USA) was affixed with an 1893 L water drum and sprayer. The other two composts included unfinished compost (from a previous windrow) and clay inputs (A), either additional landscaping residues (designated PM/A compost) or on-farm-grown legume biomass "green chop" (G) (designated PMG/A compost), and additional processing steps. The differing substrate percentages of the compost treatments and curation times was based on the direction from the compost consulting company. The landscaping residues included locally acquired grass, bush, and tree trimmings. The PM/A windrow was created on 9 June 2017 and completed on 28 July 2017. The "green-chop" included vetch and rye. The PMG/A windrow was created on 4 September 2017 and completed on 6 November 2017. Additional details regarding compost preparation are provided in [21].

The compost consulting company advised weekly windrow combining (portions of the windrow were combined with other windrow portions) and daily windrow "edge cleaning" (material around the windrow was re-incorporated into the windrow).

**Table 1.** Compost windrow substrate quantities and processing procedures. Reproduced with permission from [21].

| Treatment | Substrates and Quantities (Volume %) | Windrow Processing Procedures |
|---|---|---|
| PM | Pecan tree biomass (P): 129 m$^3$ (83%) <br> Manure (M): 27 m$^3$ (17%) | Weekly turning and watering |
| PM/A | Pecan tree biomass: 126 m$^3$ (50%) <br> Manure: 38 m$^3$ (15%) <br> Landscaping residues (A): 38 m$^3$ (15%) <br> Compost, unfinished (A): 25 m$^3$ (10%) <br> Clay (A): 25 m$^3$ (10%) | One-time clay application <br> Daily turning and watering <br> Weekly combining <br> Daily edge cleaning |
| PMG/A | Pecan tree biomass: 61 m$^3$ (31%) <br> Manure: 15 m$^3$ (8%) <br> Compost, unfinished: 15 m$^3$ (8%) <br> Clay: 15 m$^3$ (8%) <br> Green chop (G): 90 m$^3$ (46%) | One-time clay application <br> Daily turning and watering <br> Weekly combining <br> Daily edge cleaning |

Compost sub-samples were taken from each of the three composts on 15 January 2018, following US Composting Council sampling procedures [10]. Each windrow was divided into three partitions, with five sample site locations randomly generated per partition. An approximate 0.3 m by 0.3 m depth was made at each sampling point and a 2.54 cm soil core sampler was used to extrude seven samples to be composited; this totaled 35 sub-samples for each partition, for 105 samples per windrow.

### 2.3. Experimental Design and Soil Sampling Procedures

After the 2017 pecan harvest was completed, a soil amendment experiment was established in which the current in-situ application of shredded pecan biomass and manure served as the standard practice and application of the three finished composts served as the comparison treatments. The treatments were applied in a randomized complete block design to control for differences in locations, with four replications of each treatment. Approximately 18 t/ha of biomass or compost were applied in January 2018 and January 2019 and were disked in at a 15 cm depth. There was a 3.7 m buffer space between each tree and 3 m between rows. Treatments were spread 1.8 m out on either side of the trees.

Experimental site soil sampling was conducted with a 2.54 cm soil core sampler. The sampling points began at the second row from the northeast corner of the plot and two trees in (in order to provide a buffer zone). Two samples at each of the three depths (0–15 cm, 15–30 cm, 30–61 cm) were taken at each tree drip line. There was a six-tree sampling span with a buffer of two trees between each set of six trees sampled per row. A row was skipped between each row sampling point. Pre-treatment soil samples were taken in November 2017. Post-treatment soil samples were taken in August 2019.

### 2.4. Soil and Compost Characterization

Soil and compost samples were analyzed for pH, organic matter, total carbon content, nitrate-N ($NO_3$-N), K, phosphorus, Cu, Fe, Mn, Zn, electrical conductivity (EC), Na, Ca, Mg, and sodium adsorption ratio (SAR). Soil water assessment was conducted via soil gravimetric moisture content analysis. Organic matter and total carbon content were determined by the Walkley–Black method [22]. Due to the relatively high concentration of N in the compost samples compared to soils, the compost samples were analyzed at 0.1 g instead of 1 g per sample [22]. The Olsen method was used to measure available phosphorus [10,23]. A saturated paste extract (1:5 soil to deionized water ratio) was used to measure plant-available $NO_3$-N on a Technicon Autoanalyzer II, and EC, K, Ca, Na, and Mg [22,24]. Total Kjeldahl Nitrogen (TKN) was used to determine N contained in the organic substances in addition to the N contained in inorganic ammonia and ammonium ($NH_3$ and $NH_4$) [25]. Due to the relatively high N content of the compost samples, 0.5 g was used instead of the 1 g typically used for soil. The compost samples were also diluted by 50 times in deionized water prior to TKN analysis. Diethylene triamine penta-acetic acid (DTPA) extraction was used to measure available Cu, Fe, Mn, and Zn [26,27]. Composts were analyzed for phospholipid fatty acid (PLFA) by WARD Laboratories (Kearney, NE, USA) as an assessment of overall microbial communities, specifically bacterial and fungal content [28].

### 2.5. Pecan Tree Leaf Nutrition and Photosynthesis Assessments

Pecan Tree Leaf Assessments leaves were collected after the first and second year of treatment applications, in September of 2018 and 2019, and analyzed for leaf nutrients, mid-day stem water potential (SWP), and leaf soil plant analysis development (SPAD). SPAD is a measurement of plant greenness and is related to plant chlorophyll content. For sampling, six composited leaf samples were taken for each tree, avoiding newly emerged leaves and targeting the middle leaflet at random branch locations [29]. SPAD was measured using a chlorophyll meter (SPAD 502 Plus, Konica Minolta, Ramsey, NJ, USA). For nutrient content, leaves were washed with a mild detergent, rinsed with DI water, and air dried. A professional electric spice grinder with stainless steel housing and blades (Conair

Corporation, Torrington, CT, USA) was used to grind the plant leaves. Plant $NO_3$-N was analyzed by the Technicon AutoAnalyzer (Technicon, Industrial Systems, Tarrytown, NY, USA). TKN was assessed using the Technicon Block Digester, then Technicon AutoAnalyzer (Technicon Industrial Systems, Tarrytown, NY, USA). Total nutrient content was measured by microwave digestion followed by quantification using inductively coupled plasma optical emission spectroscopy (ICP-OES) (Optima 4300 DV, PerkinElmer Instrument, Norwalk, CT, USA). SWP was assessed using the Scholander Pressure Chamber instrumentation [22]. A foil bag was placed on a shaded, lower compound leaf on two opposite areas of each tree for a minimum of 15 min. Readings were taken mid-day under mostly sunny conditions, the day before the plots were irrigated.

### 2.6. Statistical Analysis

The data was analyzed using Statistical Analysis Software version 9.4 (SAS Institute Inc., Cary, NC, USA. 2016). Two treatment factors were assessed, with four levels: PM, PM/A, PMG/A, the Standard Practice, and depth with three levels: 0–15 cm, 15–30 cm, 30–61 cm. The statistical model was a randomized complete block design with repeated measurements; locations were treated as a random blocking factor. A Compound Symmetry correlation structure was assumed to account for the dependence of measurements taken on the same experimental units, at different depths. The simulated method was used to control for multiplicity on the pairwise comparisons. Analysis of variance (ANOVA) was performed on the compost treatment values to evaluate statistically significant differences in the treatments at $\alpha = 0.05$ level of significance.

## 3. Results

### 3.1. Compost Properties

The curation times for the composts were 67, 49, and 63 days for PM, PM/A, and PMG/A, respectively. The manure substrate was higher in N compared to the other N sources. The PM compost had the highest levels of TKN, phosphorus, K, Ca, and Mg, and the lowest $NO_3$-N. TKN was not significantly different among the compost treatments (Table 2). Zn and Cu differences were significant and were highest in the PM compost. Mn content was statistically significant and followed PMG/A > PM > PM/A. There were no significant differences for Fe, pH, organic carbon, or organic matter. The composts varied slightly in bulk density and were all low (0.44–0.63 g/cm³). Composts with approximately 50% moisture content average a 0.59 g/cm³ bulk density [6]. Bulk densities greater than 1.6 g/cm³ tend to restrict plant root penetration [30]. EC was significantly higher for the PM compost treatment, followed by PM/A and PMG/A. SAR and Na content were significantly different among composts; PM/A had the highest SAR and Na concentration. PM compost had the highest total microbial biomass. The higher Na and microbial biomass in PM compost were likely due to the higher fraction of the manure in the substrates prescribed by the compost consulting company (17% vs. 15% or 8%) [31] (Table 2). Walkley–Black organic carbon and TKN (instead of total elemental C and total elemental N) were used to determine final compost C/N ratios [22]. For the three composts, the C/N ratios were 11:1, 14:1, and 15:1 for the PM, PM/A, and PMG/A composts, respectively. These ratios were slightly higher than the recommended 10:1 C/N ratio for mature and stable composts [10].

**Table 2.** Properties of composts prior to application to experimental site soils. P is pecan tree biomass; M is manure; G is "green-chop" (legumes); A is additions prescribed by the compost consulting company (see Table 1). Adapted with permission from [21].

| Property | PM | PM/A | PMG/A |
|---|---|---|---|
| Total Kjeldahl N (mg/kg) | 12782 | 5508 | 7666 |
| Nitrate-N (mg/kg) | 10.45 | 813.2 | 1279 |
| Phosphorus (mg/kg) | 536.9 a | 414.9 b | 350.8 b |
| K (mg/kg) | 4722 a | 2018 b | 2365 b |

**Table 2.** *Cont.*

| Property | PM | PM/A | PMG/A |
|---|---|---|---|
| Ca (mg/kg) | 49.50 a | 34.60 b | 21.73 c |
| Mg (mg/kg) | 37.77 a | 23.67 b | 11.66 c |
| Fe (mg/kg) | 13.85 | 16.57 | 10.13 |
| Mn (mg/kg) | 24.53 b | 17.13 c | 29.07 a |
| Zn (mg/kg) | 25.29 a | 16.67 b | 20.00 b |
| Cu (mg/kg) | 3.31 a | 2.31 b | 2.24 b |
| Electrical conductivity (dS/m) | 16.93 a | 12.54 b | 11.38 b |
| Sodium adsorption ratio (meq/L) | 6.54 b | 9.02 a | 6.68 b |
| Na (meq/L) | 43.13 a | 48.77 a | 27.33 b |
| pH | 7.63 | 7.73 | 7.67 |
| Organic matter (%) | 24.30 | 13.38 | 20.31 |
| Organic carbon (%) | 14.14 f | 7.78 | 11.81 |
| Bulk density (g/cm$^3$) | 0.44 c | 0.63 a | 0.51 b |
| Microbial diversity index | 1.30 | 1.33 | 1.37 |
| Total microbial biomass | 12860 a | 4908 c | 8208 b |
| Volumetric water content (%) | 32.92 | 37.06 | 24.06 |

Values represent a mean of at least three replicates. Values in the same row followed by different letters are significantly different using the F-protected LSD level with *p*-value < 0.05.

### 3.2. Effects of Amendments on Soil Properties

Tables 3–5 show the experiment site soil properties before soil treatment application (Pre) and after two growing seasons (Post) for soil samples taken at 0–15 cm, 15–30 cm, and 30–60 cm, respectively. For many of the soil properties, there was no significant difference over time with soil treatment applications, especially for the deeper soil layers. However, some soil characteristics did show statistically significant changes.

**Table 3.** Experimental site soil properties at 0–15 cm depth of experimental site soils before (Pre) and after two years (Post) of soil treatment applications. P is pecan tree biomass; M is manure; G is "green-chop" (legumes); A represents additions prescribed by the compost consulting company (see Table 1). Standard practice is the application of shredded pecan pruning biomass and manure without composting.

| Soil Property | PM Compost Pre | PM Compost Post | PM/A Compost Pre | PM/A Compost Post | PMG/A Compost Pre | PMG/A Compost Post | Standard Practice Pre | Standard Practice Post |
|---|---|---|---|---|---|---|---|---|
| pH | 7.8 | 7.7 | 7.8 | 7.7 | 7.8 | 7.8 | 7.7 | 7.8 |
| EC (dS/m) | 0.78 | 1.68 b | 0.81 | 1.73 b | 0.73 | 1.81 b | 1.43 | 3.70 a |
| SAR | 1.62 | 1.96 | 1.63 | 2.06 | 1.63 | 1.80 | 1.42 | 1.55 |
| Na (meq/L) | 2.91 | 4.53 | 2.99 | 4.99 | 2.87 | 4.56 | 3.59 | 6.01 |
| Mg (meq/L) | 1.98 | 3.39 b | 2.45 | 3.76 b | 1.90 | 4.01 b | 2.33 | 8.93 a |
| Ca (meq/L) | 4.46 | 7.10 b | 5.96 | 8.05 b | 4.35 | 11.09 b | 6.13 | 27.0 a |
| Zn (mg/kg) | 1.98 | 2.23 a | 1.53 | 1.77 ab | 2.26 | 1.41 ab | 1.83 | 1.30 b |
| Mn (mg/kg) | 5.8 | 35 | 7.1 | 33 | 7.0 | 25 | 6.5 | 26 |
| Fe (mg/kg) | 3.8 | 33.7 | 5.8 | 29.6 | 5.0 | 19.5 | 6.3 | 25.9 |
| Cu (mg/kg) | 0.71 | 1.21 | 0.78 | 1.31 | 0.91 | 1.22 | 0.81 | 1.30 |
| Organic matter (%) | 1.75 | 2.14 a | 1.46 | 2.04 ab | 1.51 | 1.86 ab | 1.58 | 1.36 b |
| Organic carbon (%) | 1.02 | 1.24 a | 0.85 | 1.18 a | 0.88 | 1.08 ab | 0.92 | 0.70 b |
| TKN (mg/kg) | 1031 | 1181 a | 1026 | 1007 ab | 869 | 863 b | 913 | 883 b |
| Nitrate-N (mg/kg) | 2.5 | 8.2 | 2.4 | 7.3 | 3.1 | 8.1 | 5.1 | 6.7 |
| Phosphorus (mg/kg) | 5.5 | 25.8 a | 7.2 | 13.3 b | 8.3 | 18.5 ab | 10.3 | 23.6 a |
| K (mg/kg) | 30 | 51 | 30 | 53 | 34 | 86 | 49 | 200 |

Values represent the mean across the four blocks. Values in the same row followed by different letters indicate that the Post value was significantly different from the Pre value using the F-protected LSD level with *p*-value < 0.05. EC is electrical conductivity, SAR is sodium adsorption ratio, TKN is total Kjeldahl nitrogen.

**Table 4.** Experimental site soil properties at 15–30 cm depth of experimental site soils before (Pre) and after two years (Post) of soil treatment applications. P is pecan tree biomass; M is manure; G is "green-chop" (legumes); A represents additions prescribed by the compost consulting company (see Table 1). Standard practice is the application of shredded pecan pruning biomass and manure without composting.

| Soil Property | PM Compost Pre | PM Compost Post | PM/A Compost Pre | PM/A Compost Post | PMG/A Compost Pre | PMG/A Compost Post | Standard Practice Pre | Standard Practice Post |
|---|---|---|---|---|---|---|---|---|
| pH | 7.7 | 7.7 | 7.7 | 7.7 | 7.8 | 7.8 | 7.7 | 7.7 |
| EC (dS/m) | 1.16 | 1.62 | 1.34 | 1.51 | 1.2 | 1.61 | 1.94 | 2.83 |
| SAR | 4.48 | 4.41 | 4.87 | 4.61 | 4.61 | 4.41 | 5.36 | 6.4 |
| Na (meq/L) | 2.05 | 1.96 | 2.07 | 2.04 | 2.08 | 1.92 | 1.96 | 1.97 |
| Mg (meq/L) | 0.85 | 0.86 | 0.92 | 0.71 | 0.78 | 0.59 | 1.15 | 0.62 |
| Ca (meq/L) | 5.8 | 43 a | 5.2 | 42 a | 5.6 | 28 b | 5.5 | 28 b |
| Zn (mg/kg) | 2.69 | 3.09 | 4.19 | 3.13 | 2.8 | 3.07 | 2.87 | 6.04 |
| Mn (mg/kg) | 6.9 | 7.1 | 11 | 7.1 | 7.0 | 8.0 | 8.2 | 16.9 |
| Fe (mg/kg) | 3.5 | 43.5 | 3.9 | 43.3 | 4.0 | 25 | 4.9 | 25.9 |
| Cu (mg/kg) | 0.84 | 1.15 | 0.94 | 1.17 | 0.92 | 1.16 | 0.97 | 1.26 |
| Organic matter (%) | 1.14 | 1.52 | 0.86 | 0.91 | 1.04 | 1.25 | 1.06 | 1.01 |
| Organic carbon (%) | 0.66 | 0.88 | 0.50 | 0.53 | 0.60 | 0.72 | 0.62 | 0.51 |
| TKN (mg/kg) | 619 | 932 | 555 | 734 | 619 | 758 | 596 | 673 |
| Nitrate-N (mg/kg) | 3.4 | 5.2 | 3.2 | 5.1 | 3.0 | 4.4 | 2.8 | 5.0 |
| Phosphorus (mg/kg) | 1.8 | 12.0 ab | 2.43 | 7.3 b | 4.8 | 10.0 ab | 4.0 | 13.4 a |
| K (mg/kg) | 35 | 46 | 45 | 33 | 48 | 76 | 61 | 191 |

Values represent the mean across the four blocks. Values in the same row followed by different letters indicate that the Post value was significantly different from the Pre value using the F-protected LSD level with *p*-value < 0.05. EC is electrical conductivity, SAR is sodium adsorption ratio, TKN is total Kjeldahl nitrogen.

For the top soil layer (0–15 cm), all soil treatments significantly increased the soil electrical conductivity and available phosphorus, Mg, and Ca concentrations (Table 3); these increases were attributed to the salt and nutrient content of the dairy manure. Slight increases in soil Na content and SAR (Figure 1) were also observed, but these were generally not statistically significant, suggesting that the added salts in the soil treatments were not of immediate concern for soil salinity or sodicity but may require monitoring, especially for the standard practice treatment, which has a greater fraction of manure than the PM/A or PMG/A composts, and does not experience any of the windrow processes steps that could leach out some of the salts prior to soil application.

The addition of soil treatments generally increased the available micronutrients (Zn, Mn, Fe and Cu); only the changes in Zn were statistically significant between pre and post measurements for a given treatment. A decrease in Zn was noted for the PMG/A compost and standard practice treatments compared to the increase in soil Zn for the PM and PM/A composts; the reason for the decrease was not immediately clear. Across all samples, there were some small but statistically significant differences in available soil Fe and Cu (Figure 2). Soil organic carbon and organic matter significantly increased with compost additions but showed a decrease for the standard practice treatment. TKN showed a statistically significant increase with the PM compost but a decrease with the other three treatments (Figure 2).

At the lower soil depths (15–30 cm and 30–60 cm), changes in soil properties were generally small (Tables 4 and 5). There were increases in Ca content in both lower layers for all soil treatments, although only the increases in the 15–30 cm depth layer were statistically significant (Table 4). There were statistically significant increases in both lower layers for all soil treatments for available phosphorus. Combined, this indicates that both Ca and phosphorus from the soil treatments were at least somewhat mobile in the soil profile. In

neutral or calcareous soils, inorganic phosphorus precipitates as calcium phosphate or is adsorbed to organic matter and clay particle surfaces [25].

**Table 5.** Experimental site soil properties at 30–60 cm depth of experimental site soils before (Pre) and after two years (Post) of soil treatment applications. P is pecan tree biomass; M is manure; G is "green-chop" (legumes); A represents additions prescribed by the compost consulting company (see Table 1). Standard Practice is the application of shredded pecan pruning biomass and manure without composting.

| Soil Property | PM Compost Pre | PM Compost Post | PM/A Compost Pre | PM/A Compost Post | PMG/A Compost Pre | PMG/A Compost Post | Standard Practice Pre | Standard Practice Post |
|---|---|---|---|---|---|---|---|---|
| pH | 7.8 | 7.8 | 7.7 | 7.8 | 7.8 | 7.8 | 7.7 | 7.8 |
| EC (dS/m) | 1.73 | 1.61 | 1.76 | 1.54 | 2.11 | 1.92 | 2.26 | 2.70 |
| SAR | 5.81 | 4.34 | 6.25 | 4.81 | 7.06 | 5.19 | 6.92 | 6.56 |
| Na (meq/L) | 2.14 | 1.96 | 2.14 | 2.09 | 2.43 | 1.96 | 2.18 | 2.17 |
| Mg (meq/L) | 0.54 | 0.75 | 0.45 | 0.35 | 0.76 | 0.35 | 1.00 | 0.38 |
| Ca (meq/L) | 4.1 | 37 | 3.5 | 32 | 4.0 | 30 | 4.0 | 30 |
| Zn (mg/kg) | 4.07 | 2.92 | 6.03 | 3.11 | 4.75 | 3.92 | 4.14 | 5.18 |
| Mn (mg/kg) | 10.6 | 6.9 | 15.4 | 7.5 | 11.7 | 10.7 | 11.3 | 14.2 |
| Fe (mg/kg) | 3.5 | 41.1 | 3.0 | 36.2 | 3.7 | 30.4 | 3.7 | 35.7 |
| Cu (mg/kg) | 0.83 | 1.18 | 0.75 | 1.10 | 0.87 | 1.20 | 0.87 | 1.21 |
| Organic matter (%) | 0.86 | 1.11 | 0.69 | 0.63 | 0.87 | 0.71 | 1.04 | 0.62 |
| Organic carbon (%) | 0.50 | 0.64 | 0.40 | 0.37 | 0.50 | 0.41 | 0.61 | 0.31 |
| TKN (mg/kg) | 474 | 707 | 446 | 598 | 463 | 548 | 461 | 624 |
| Nitrate-N (mg/kg) | 5.5 | 5.0 | 5.4 | 4.7 | 5.4 | 3.9 | 4.8 | 5.1 |
| Phosphorus (mg/kg) | 3.1 | 11.0 a | 2.4 | 5.3 b | 5.0 | 7.3 ab | 4.0 | 10.2 ab |
| K (mg/kg) | 29 | 42 | 42 | 25 | 54 | 78 | 79 | 204 |

Values represent the mean across the four blocks. Values in the same row followed by different letters indicate that the Post value was significantly different from the Pre value using the F-protected LSD level with *p*-value < 0.05. EC is electrical conductivity, SAR is sodium adsorption ratio, TKN is total Kjeldahl nitrogen.

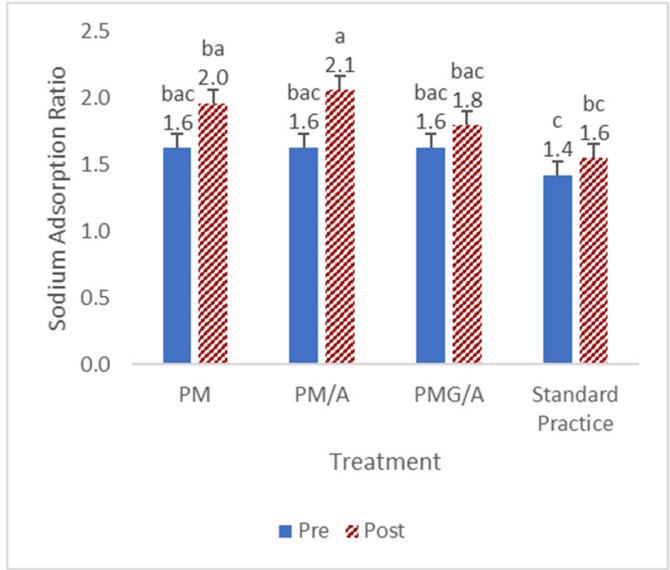

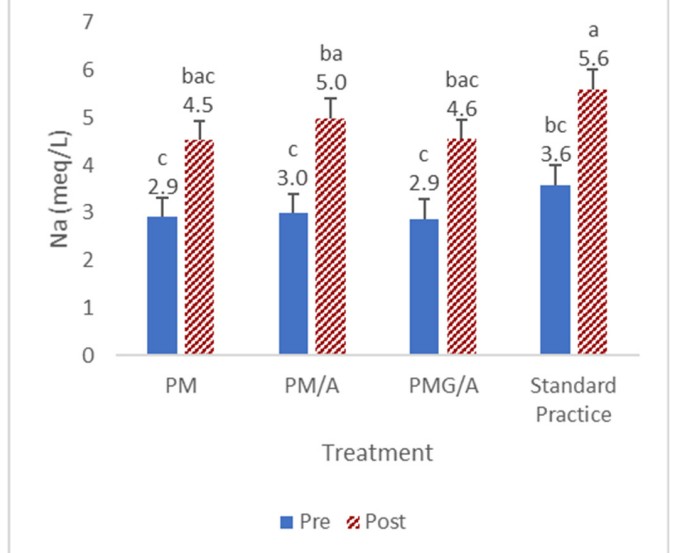

**Figure 1.** Soil sodium adsorption ratio (**left**) and sodium content (**right**) at 0–15 cm depth of experimental site soils before (Pre) and after two years (Post) of soil treatment applications. P is pecan tree biomass; M is manure; G is "green-chop" (legumes); A represents additions prescribed by the compost consulting company (see Table 1). Standard practice is the application of shredded pecan pruning biomass and manure without composting. Different letters indicate statistical significance between all treatments and times.

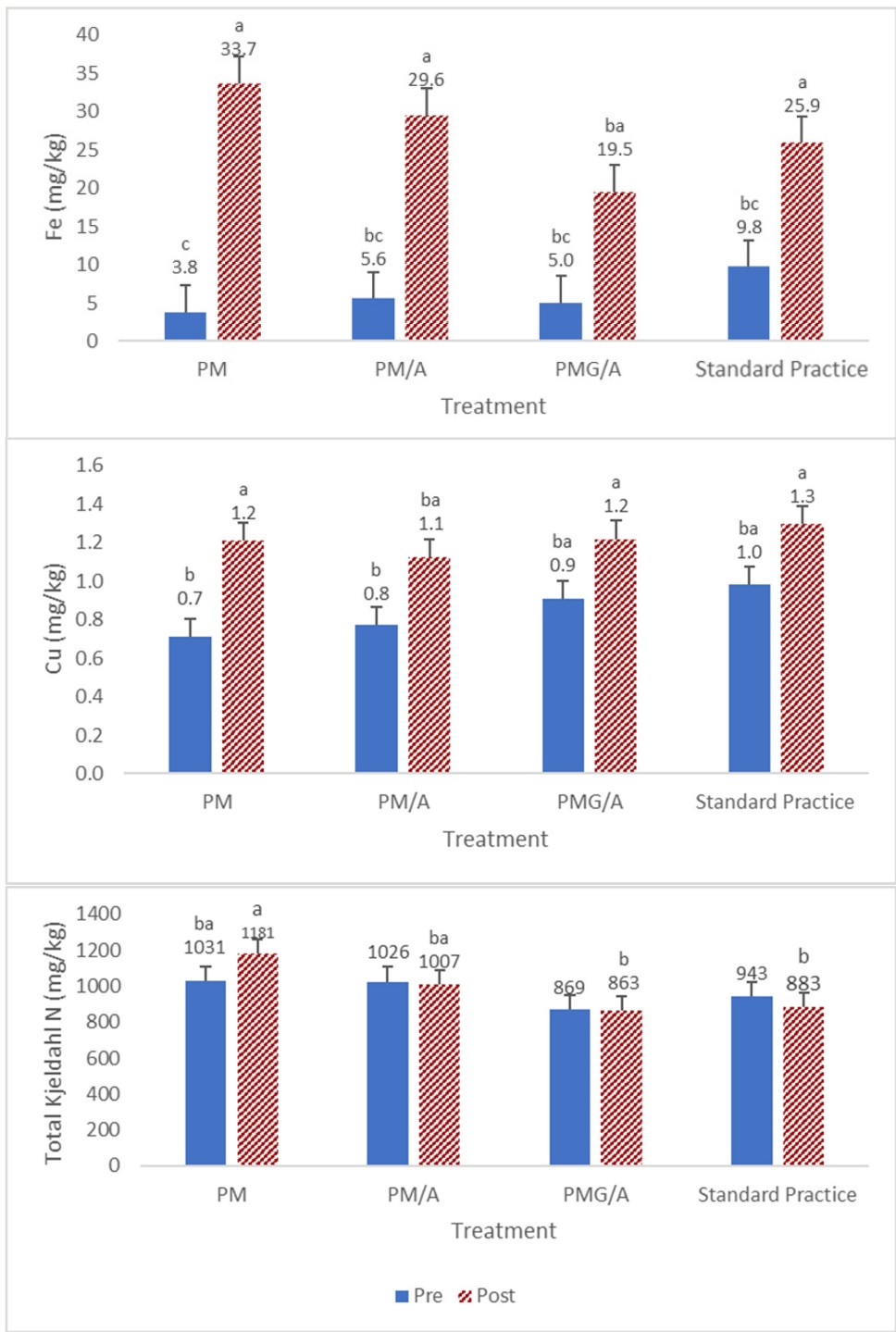

**Figure 2.** Soil nutrient content: iron (**top**), copper (**middle**), and total Kjeldahl nitrogen (**bottom**) at 0–15 cm depth of experimental site soils before (Pre) and after two years (Post) of soil treatment applications. P is pecan tree biomass; M is manure; G is "green-chop" (legumes); A represents additions prescribed by the compost consulting company (see Table 1). Standard practice is the application of shredded pecan pruning biomass and manure without composting. Different letters indicate statistical significance between all treatments and times.

Available TKN and Fe and Cu contents increased in the other two layers for all soil treatments, although the increases were not significant. Unlike in the top layer, increases in available Mn and Zn were not observed. The post soil K content for all three soil layers was noticeably higher for the standard practice treatment. The soil, a Reakor loam with low over-

all organic matter and high sand content, is expected to have relatively high translocation of positively charged ions to lower depths under the floor irrigation conditions.

### 3.3. Effects of Amendments on Plant Properties

At one year and two years post-treatment, no statistically significant differences were observed in pecan leaf parameters between treatments or between years (Table 6). More differences may have been apparent if pre-treatment samples were compared with post-treatment samples. At the times of sampling, plant leaf contents of phosphorus and K were below an "optimal" range [29] for pecan trees (Table 7). This suggests that all of the trees may benefit from additional uptake of these nutrients but that there was no observable effect of the soil amendment treatments on leaf nutrient content.

**Table 6.** Pecan leaf properties during the first and second years after soil amendment treatment applications. P is pecan tree biomass; M is manure; G is "green-chop"; A is additions (see Table 1). Standard practice is the application of shredded pecan pruning biomass and manure without composting.

| Leaf Property | PM Compost Year 1 | PM Compost Year 2 | PM/A Compost Year 1 | PM/A Compost Year 2 | PMG/A Compost Year 1 | PMG/A Compost Year 2 | Standard Practice Year 1 | Standard Practice Year 2 |
|---|---|---|---|---|---|---|---|---|
| SWP (bar) | 8.2 | 9.6 | 7.8 | 9.8 | 8.4 | 10.0 | 8.4 | 8.9 |
| SPAD (nm) | 46.8 | 47.0 | 46.6 | 47.3 | 46.7 | 46.4 | 46.3 | 47.4 |
| Fe (mg/kg) | 787 | 80 | 82 | 86 | 115 | 87 | 496 | 76 |
| Mn (mg/kg) | 79 | 94 | 72 | 97 | 69 | 83 | 76 | 95 |
| Ni (mg/kg) | 1.4 | 1.5 | 1.2 | 1.3 | 1.0 | 1.4 | 1.4 | 1.6 |
| Zn (mg/kg) | 14 | 43 | 16 | 49 | 29 | 49 | 14 | 40 |
| Phosphorus (mg/kg) | 1054 | 1070 | 1067 | 1050 | 1079 | 1015 | 1106 | 1075 |
| K (mg/kg) | 8264 | 8890 | 8515 | 9046 | 9058 | 8677 | 8729 | 9393 |
| Na (mg/kg) | 84 | 74 | 62 | 122 | 62 | 103 | 86 | 102 |
| TKN (%) | 2.2 | 1.8 | 2.3 | 2.0 | 2.2 | 1.9 | 2.3 | 2.2 |

SWP is mid-day stem water potential; SPAD is soil plant analysis development; TKN is Total Kjeldahl N. SWP and SPAD were measured on a fresh leaf basis; all other variables were assessed on a leaf dry weight basis.

**Table 7.** Phosphorus and K "optimal" nutrient ranges in pecan leaves [29], assessed on a dry leaf weight basis, as compared to soil-incorporated treatment pecan leaf ranges. P is pecan tree biomass; M is manure; G is "green-chop"; A is additions (see Table 1). Standard practice is the application of shredded pecan pruning biomass and manure without composting.

| | Phosphorus (mg/kg) | K (mg/kg) |
|---|---|---|
| "Optimal" Ranges | 1400–1900 | 12,000–25,000 |
| PM Compost | 1050–1070 | 8300–8900 |
| PM/A Compost | 1050–1070 | 8500–9000 |
| PMG/A Compost | 1020–1080 | 8600–9000 |
| Standard Practice | 1070–1100 | 8700–9400 |

## 4. Discussion

### 4.1. Soil Amendment Quality Impacts and Decision Factors

A key decision for soil quality and nutrient management is the choice to use or not to use available organic materials such as pecan tree trimmings and dairy manure. Low soil fertility is a primary hindrance in agricultural production and associated profits for farmers [32]. High quality soil, assessed by physical, chemical, and biological characteristics, improves the interconnected water, air, and plant systems and positively impacts crop production [13,15]. In-situ compost application has been shown to provide crop nutrient and water conservation benefits, which relate to economic benefits associated with compost nutrients as a potential supplement for synthetic fertilizers [33]. If farmers continue incorporation of organic materials into orchard soils, they may be able to reduce their long-term fertilizer and irrigation needs and associated costs. In one compost application study, compost application enabled a 36% saving in urea and completely replaced

phosphorus and K fertilizers [34]. The compost substrates used here translated to higher nitrate, Fe, and Cu soil nutrient content, particularly in the top 0–15 cm soil depths where changes in soil chemistry and nutrition often occur [28]. Increasing nutrient availability is particularly important in soil-depleted arid regions where higher pH, limited water, and deficits in some plant macronutrients, such as phosphorus, are challenges [35]. With several applications, compost-amended soils can increase soil water and nutrient retention and reduce associated nutrient and water inputs [36].

Once the decision has been made to use available organic materials, additional decisions must be made about the return on investment for amendment preparation steps (here, shredding and mixing vs. composting) and application costs. Composting regimes can become quite complex, with substantial material and labor inputs (as were used for the production of the PM/A and PMG/A composts), with few apparent benefits compared to not composting at all or using simpler composting production methods [21]. Farmers should be provided accurate product and processing information so that the costs and benefits can be assessed. Such information is likely to require cooperation between academic and extension resources with organic waste producers and farmers in order to create a viable organic waste-to-resource program.

As an environmental best management practice and climate change mitigation tactic, incorporating high-C organic materials is expected to increase the soil carbon content. In this study, there were no significant differences amongst treatments for SOM or organic carbon content, however, slight trends upwards were seen in the top soil layers for the composted treatment but not the un-composted standard practice treatment. Future research is needed in terms of on-farm carbon inventory: compost carbon dioxide emissions, soil amendment carbon inputs, soil and plant carbon sequestration, and plant carbon dioxide emissions.

*4.2. Limitations of Short-Term Studies and Leaf Measurements for Orchard Management Decision-Making*

Research on methods for safe, feasible, and economically viable utilization of common organic waste streams is limited [1,35]. Such research is especially challenging given the number of factors involved (soil, climate, crop, etc.) and the need for multi-year or multi-decade time scales to observe the most meaningful responses. A longer-term study of compost-amended soils may show greater effects over time for nutrient availability, the soil humus fraction, and soil and plant water statuses without incorporating additional soil amendments.

In the short-term, demonstration of "doing no harm" may be sufficient relative to the benefits of organic waste management. For pecan trees grown on alkaline, irrigated arid-region soils, the key criteria to be met are related to soil pH, salinity, and available nutrients. Here, pH remained approximately the same for all treatments over two years. Soil salinity can adversely affect pecan production [34]. Here, the soil electrical conductively and sodium adsorption ratio did show some concerning trends upwards for the top soil layer, indicating that close monitoring is needed in the long-term. Use of dairy manure, therefore, may be limited in this region, with or without composting. For available nutrients, the goals are to avoid nutrient immobilization from application of too much available carbon or element toxicity from over-application. Composting the substrates before land application to reduce this concern is likely to be an important factor in favor of adopting a pretreatment process. Here, the high-C pecan trimming biomass was able to be safely degraded via microbial action with the manure and plant-based, higher-N compost substrates [37]. Microbial decomposition can be hindered by a lack of microorganisms in the initial substrate material; this can translate to reduced soil benefits, including less microbial abundance and plant nutrient facilitation [3,7]. Over time, soil respiring and metabolizing microorganisms, including arbuscular mycorrhizal, actinomycetes, and protozoa, convert organic compounds to plant-usable inorganic compounds [3,38]. Total microbial analyses of the composts used in this study showed that the PM compost had

the highest abundance of Gram-positive actinomycete, Gram-negative rhizobia, and total fungi, namely arbuscular mycorrhizal and saprophyte abundance (Table 2), suggesting that complex composting practices are not needed to achieve the soil microbial benefits of composting [21].

## 5. Conclusions

For most soil properties, the soils amended with the three compost treatments were not significantly different than the soil treated with the standard practice of applying uncomposted pecan residues and dairy manure. The content of several nutrients, however, did increase from before application to two years after soil treatment application for all treatments within the 0–15 cm soil layer. The composts produced with additional inputs and more complex methods, PM/A and PMG/A, did not result in increased soil or plant benefits over the time period studied. Therefore, PM compost processing may be the least costly and labor-intensive option for combined waste management and crop production without substantially higher risk of soil or crop contamination. Longer-term studies, especially those that include pecan production yields and quality and soil carbon balance monitoring, are needed to more completely evaluate the trade-offs between schemes for utilizing local organic wastes in crop production.

**Author Contributions:** Conceptualization, E.F.C., R.J.H. and M.D.; Methodology, E.F.C., R.F., R.J.H. and C.V.-C.; Validation, R.J.H. and M.D.; Formal analysis, E.F.C., R.J.H. and C.V.-C.; Investigation, E.F.C.; Resources, R.F., C.E.B. and C.V.-C.; Data curation, R.F.; Writing—review & editing, E.F.C. and C.E.B.; Supervision, R.F. and C.E.B.; Project administration, R.F., C.E.B. and R.J.H.; Funding acquisition, R.F., C.E.B. and M.D. All authors have read and agreed to the published version of the manuscript.

**Funding:** This research was funded by the U.S. Department of Agriculture National Needs Fellowship grant number [2015-38420-23706].

**Data Availability Statement:** The data presented in this study are available on request from the corresponding author.

**Conflicts of Interest:** The authors declare no conflict of interest.

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
