# Peer review of "Pecan Biomass and Dairy Manure Utilization: Compost Treatment and Soil In-Situ Comparisons of Selected Pecan Crop and Soil Variables"

_processes, doi:10.3390/pr11072046_

Round 1
Reviewer 1 Report
The article deals with an interesting and important issue, however, the reviewer has some important remarks:
- Table 1 does not show compost production methods as stated in lines 120-121
- Why is the C/N ratio of the composts not shown? The C/N ratio is a commonly used method to assess compost maturity/stabilization. Therefore, it is not known whether the composts used were actually composts.
- Please specify the qualitative composition of "landscaping residues" - studies must be able to be replicated by other scientists.
- What did the authors mean by using in groups PM/A and PMG/A - compost, unfinished?
- The structure of the substrates seems to be chaotic - please justify the percentage of the initial biomass used in the experimental groups.
Author Response
Full response to reviewers and updated manuscript is attached.

Reviewer 2 Report
All comments are listed in the attached file.

All comments are listed in the attached file.
Round 2
Reviewer 2 Report
All my comments have been answered and the paper can be published.